# Age, Growth, and Validation of Otolith Morphometrics as Predictors of Age in the Blackspot Seabream, *Pagellus bogaraveo*, (Brunnich, 1768) from the Eastern Adriatic Sea

**Antonela Paladin [1],***, **Nika Ugrin [2]**, **Sanja Matić-Skoko [3]**, **Branko Dragičević [3]** and **Jakov Dulčić [3]**

1   Faculty of Science, University of Split, Ruđera Boškovića 33, 21 000 Split, Croatia
2   Department of Marine Studies, University of Split, Ruđera Boškovića 37, 21 000 Split, Croatia; nugrin@unist.hr
3   Institute of Oceanography and Fisheries, Šetalište Ivana Meštrovića 63, 21 000 Split, Croatia; sanja@izor.hr (S.M.-S.); brankod@izor.hr (B.D.); dulcic@izor.hr (J.D.)
*   Correspondence: apaladin@pmfst.hr

**Abstract:** The age and growth of the blackspot seabream, *Pagellus bogaraveo*, were determined by examining sagittal otoliths from fish sampled in the eastern Adriatic Sea. A total of 674 specimens (181 males, 90 hermaphrodites, 108 females, and 295 immatures) ranging from 8.80 to 47.27 cm ($21.93 \pm 9.00$ cm) in total length were analyzed. The maximum observed age for the whole sample was 13 years. The sample was dominated by 1-year-old specimens as a consequence of an abundance of juvenile specimens collected by beach seines. Growth was described by the von Bertalanffy growth curve ($L_\infty$ = 52.3 cm, K = 0.15 year$^{-1}$, $t_0$ = 0.49 year, $R^2$ = 0.97), and the growth performance index ($\Phi'$) was 2.61. The length, width, thickness, and mass of the otoliths were compared with the total length and age of *P. bogaraveo* from the eastern Adriatic. The analysis showed that the measures were adequate predictors of age. These results can be of value for more effective management measures aimed at the conservation of this species.

**Keywords:** Sparidae; age determination; Mediterranean Sea; hermaphroditism; otolith morphometry

**Key Contribution:** This study brings the first data on the demographic structure of the blackspot seabream, *Pagellus bogaraveo*, in the Adriatic Sea. The classical method of age determination by counting the rings in the sagittal otoliths shows that the maximum observed age of this species was 13 years, while the relationship between the otolith dimensions and the age of the fish shows that the age of this species can be best estimated by the length of the otolith. The obtained results can have high applicability for national and regional fisheries' management purposes and can contribute to research that requires a rapid assessment of fish population age structure.

## 1. Introduction

The Sparidae family includes 35 genera and 112 species occurring in all the world's oceans [1]. In the Mediterranean Sea, the family is represented by 11 genera and 23 species, and in the Adriatic, it is represented by 10 genera and 18 species [2,3]. Among these, three species of genus Pagellus are present in the Adriatic Sea, namely *P. erythrinus* (Linnaeus, 1758) (common pandora), *P. acarne* (Risso, 1827) (axillary seabream), and *P. bogaraveo* (Brünnich, 1768) (blackspot seabream). Species of the Sparidae family are mainly benthopelagic species that live at depths of up to 500 m, but more commonly they live at depths of up to 150 m [4].

The blackspot seabream, *Pagellus bogaraveo,* is distributed in the eastern Atlantic, from the southern part of Norway to Cape Blanc, Madeira, and the Canary Islands, and in the central and western Mediterranean Sea [2]. In the Adriatic Sea, it is mainly present in the central and south Adriatic [5,6]. *P. bogaraveo* can grow up to 80 cm in length [2]. It is

a sequential protoandric hermaphrodite [7]. As an important commercial species in the Mediterranean Sea, it is mainly caught by trawls, traps, longlines, and hooks [2]. The prescribed MLS (minimum landing size) for *P. bogaraveo* for the EU (EC, 2019) and the Mediterranean Sea (EC, 2006) is 33 cm [8,9]. In the Mediterranean, the genus *Pagellus* represents appreciated fishery resources and occurs throughout the basin, although with differences in frequency of occurrence and relative abundance between the western and eastern regions [10]. According to the data from the MEDITS fisheries biology expedition from 1996 to 2010, the blackspot seabream was mainly recorded in the central Adriatic Sea. The highest abundance was found in the shallow waters in the eastern Adriatic. The frequency of occurrence was low and fluctuated around a mean value of 9% [11]. For the Mediterranean population, some characteristics such as the length–weight relationship, e.g., [12–14], age and growth, e.g., [12,14–20], reproduction [14,21,22], diet, e.g., [23,24], and otolith morphology [25] have been studied. Age and growth have been widely studied in the Mediterranean and Atlantic areas, but there is a lack of such data for the Adriatic Sea. The largest age range was recorded by Gueguen [16] in the Bay of Biscay (1–20 years). In other research areas, the age range was from 1 to 14 years [15,17,18]. The smallest age range (0–3) was recorded by Mytilineou and Papaconstantinou [12] in the Aegean Sea. All authors determined age by counting rings on sagittal otoliths.

However, biological information regarding this species in the Adriatic Sea is poorly known. Županović and Jardas [6] consider this species to be relatively rare, mostly inhabiting larger depths of up to at least 300 m, while juveniles can be found closer to the coast. This study is the first attempt to investigate the age and growth of the blackspot seabream, *P. bogaraveo,* from the eastern Adriatic population, with the aim that the obtained data can be used in the management of this species in the Adriatic Sea. The fish age was determined by counting rings in sagittal otoliths, which is one of the most time-consuming aging methods. The utility of otolith morphometrics is evaluated (length, width, thickness, and mass) as an alternative. Compared with conventional age determination, methods that use otolith size are less time-consuming and have the potential to produce a high number of observations in a short time.

## 2. Materials and Methods

### 2.1. Sampling

Samples of the blackspot seabream were obtained monthly between October 2007 and August 2009. The samples originated from commercial landings and catches of demersal longlines and beach seine "migavica" operating in the eastern Adriatic Sea, respectively. Samples of the former were obtained with the help of fish market staff who assisted us with infrequent landings of blackspot seabream, while samples of the latter were obtained through onboard sampling during a fishery monitoring program performed by the Institute of Oceanography and Fisheries in Split, Croatia. These types of equipment target different components of the blackspot seabream population, where longlines effectively catch adults, while beach seines catch juveniles, which represent a discard for that type of fishery. The obtained samples were subsequently analyzed in the laboratory. Each fish was measured to the nearest 0.1 cm total length (TL) and weighed (W) to the nearest 0.1 g. Sex was determined by macroscopic analysis of the gonads, and sagittal otolith pairs were removed, cleaned, and stored in Eppendorf tubes for future analyses.

### 2.2. Age and Growth Estimation

To determine the age composition of the blackspot seabream population, the method of direct reading of the otoliths by counting annual growth rings on the sagittal otoliths was used. After the opaque zone, hyaline and opaque rings alternated. One hyaline and one opaque ring indicate one year of age. Age was read from polished otoliths observed under the reflective light of a Leica MZ75 stereomicroscope on a black background. All otoliths were photographed with a Leica D-LUX 3 digital camera. Age was read three times to avoid subjectivity in the reading. Based on age data, length growth was calculated using

Von Bertalanffy's growth equation [26] according to the formula: $TL = L_\infty [1 - e^{-K(t - t_0)}]$, where TL—total length at time t, $L_\infty$—asymptotic value of length TL, K—growth coefficient, and $t_0$—theoretical age of the fish at length $L_0$.

### 2.3. Otolith Morphometrics

The length, width, and thickness of the otoliths in the selected subsample were measured with a caliper with an accuracy of 0.01 mm, and the mass of the otoliths with an analytical balance was measured with an accuracy of 0.01 g. The length of the otoliths was measured along the longest axis, and the width was measured perpendicular to the length of the otolith. The terminology proposed by Tuset et al. [27] was used to describe the otolith morphology. Relationships between fish age and total length and otolith morphometrics (length, width, thickness, and mass) were constructed using the linear model.

### 2.4. Statistical Analysis

Monthly sampling data were stored in a database in Microsoft Excel 2016, MSO, Microsoft Corporation, Redmond, WA, USA. The same program was also used to calculate mean values, standard deviation, and minimum and maximum values. Statistical analyses were performed in the R software, version R 4.2.2. PBC, Boston, MA, USA. The sex ratio was tested for significance (female to male = 1:1) using the chi-squared test ($p < 0.05$). The normality of the distribution was tested by the two-sample Kolmogorov–Smirnov test ($p < 0.01$). Differences in length and weight frequencies between sexes were tested by the *t*-test. Growth parameters from von Bartalanffy's formula were calculated in the software package Statistica, version 8, StatSoft, Inc., Tulsa, OK, USA. The parameters were calculated using user-specified regression by least-squares estimation. The index of growth performance ($\Phi'$) was calculated in order to allow a more meaningful comparison with other studies [28]. $\Phi' = \log K + 2 \log L_\infty$, and the age at which blackspot seabream reached 95% of the asymptotic value of length Lt ($A_{0.95}$) was determined according to the formula [29] $A_{0.95} = t_0 + 2.996/K$, where $t_0$, K, and $L_\infty$ are parameters from von Bertalanffy's growth equation. The multivariate Hotelling $T^2$-test was used to compare growth parameters between sexes [30]. Statistical differences in morphometry between left and right otoliths were tested by the *t*-test ($p \leq 0.05$).

## 3. Results

In total, 674 individuals from the eastern Adriatic were analyzed. The total length range was from 8.80 to 47.80 cm (21.93 ± 9.000 cm) (Figure 1). The sample consisted of 181 males (32.67%), 90 hermaphrodites (16.2%), 108 females (19.49%), and 295 immatures (31.59%). The total length ranges of females and males were from 22.80 to 47.80 cm (33.80 ± 4.30 cm) and from 18.00 to 42.50 cm (26.07 ± 4.470 cm), respectively. The total length range of hermaphrodites was from 24.30 cm to 32.50 cm (28.74 ± 1.830 cm) (Figure 2), and that of immature individuals was from 8.80 to 20 cm (14.62 ± 3.500 cm). The mean values of the total lengths of males and females were significantly different (*t*-test; $p < 0.05$). The mean value of total lengths of hermaphrodites compared to males and females was also statistically significant (Kolmogorov–Smirnov test; $p < 0.01$). The weight range of the total sample was from 4.44 g to 1495 g (245.27 ± 552.900 g). The weight range of males was from 74.87 to 1069.11 g, the weight range of females was from 172.34 to 1495 g, and the weight range of hermaphrodites was from 194 to 495 g. The mean value of the weight of males was significantly lower than the mean value of the weight of females (*t*-test; $p < 0.05$). The weight of females and males was statistically significantly different from the weight of hermaphrodites (Kolmogorov–Smirnov test; $p < 0.01$). The ratio of females to males within the population of the blackspot seabream in the eastern Adriatic was 1:0.81 with a male dominance. The sex ratio was statistically significant (Chi-square test; $p < 0.05$).

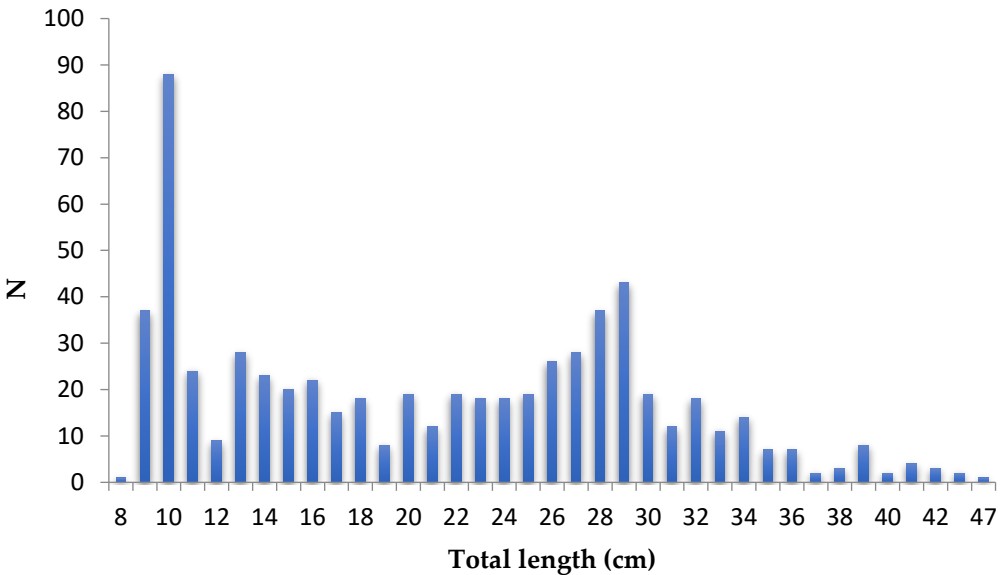

**Figure 1.** Length frequency distributions of total sample of the blackspot seabream, *Pagellus bogaraveo*, in the eastern Adriatic Sea.

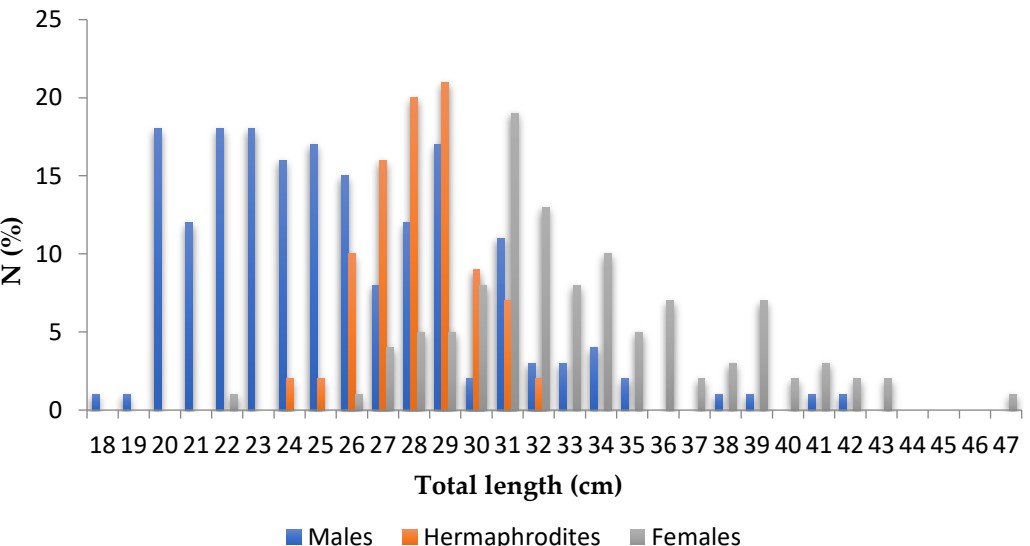

**Figure 2.** Length frequency distributions of the blackspot seabream, *Pagellus bogaraveo*, females, males, and hermaphrodites in the eastern Adriatic Sea.

The otolith of the blackspot seabream contained a wide opaque zone in the center. The width of the opaque and hyaline rings was similar except at the edges of the otolith where the rings narrowed (Figure 3).

In the total sample, the most represented individuals were one-year-old, i.e., juvenile, individuals (N = 182), i.e., 32.9%. The oldest female was 13 years old, and the youngest female was 4 years old. The oldest male was 12 years old, and the youngest was 3 years old. Hermaphrodites were recorded in the age range of 3 to 6 years. The eastern Adriatic blackspot seabream population was dominated by females aged from 5 to 7, males aged from 3 to 4, and juveniles aged one year (Figure 4). Females older than 8 years and males older than 7 years were poorly represented in the population (Table 1). Males were more represented in the age groups of 3 and 4 years, and females were more represented in the age groups from 5 to 13 years. The growth of the blackspot seabream was described by the von Bertalanffy growth equation, and growth curves were obtained for males, females, and the total sample (Figure 5).

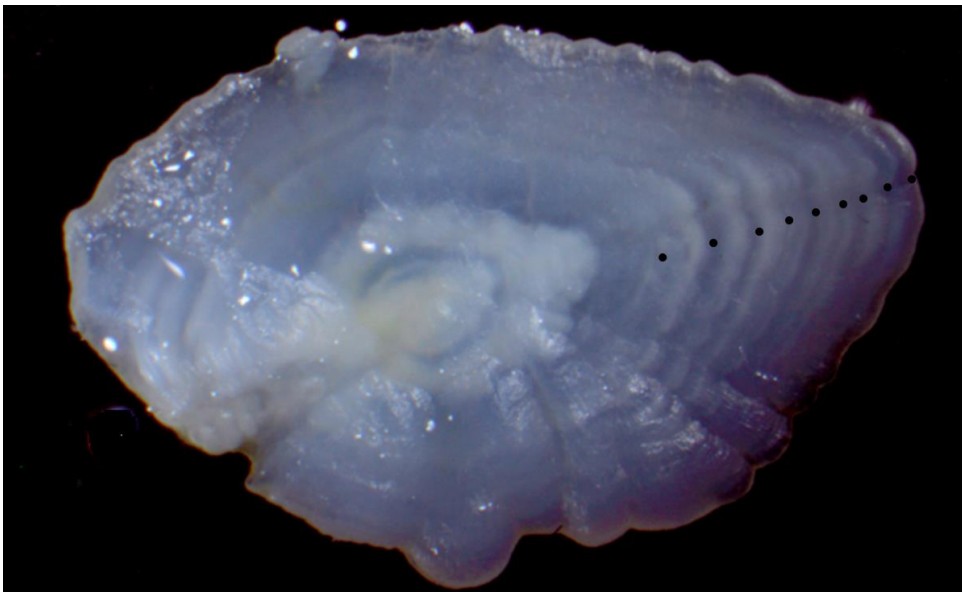

**Figure 3.** Sagittal otolith of a 9-year-old *P. bogaraveo* in the eastern Adriatic Sea. The black dots indicate the growth rings.

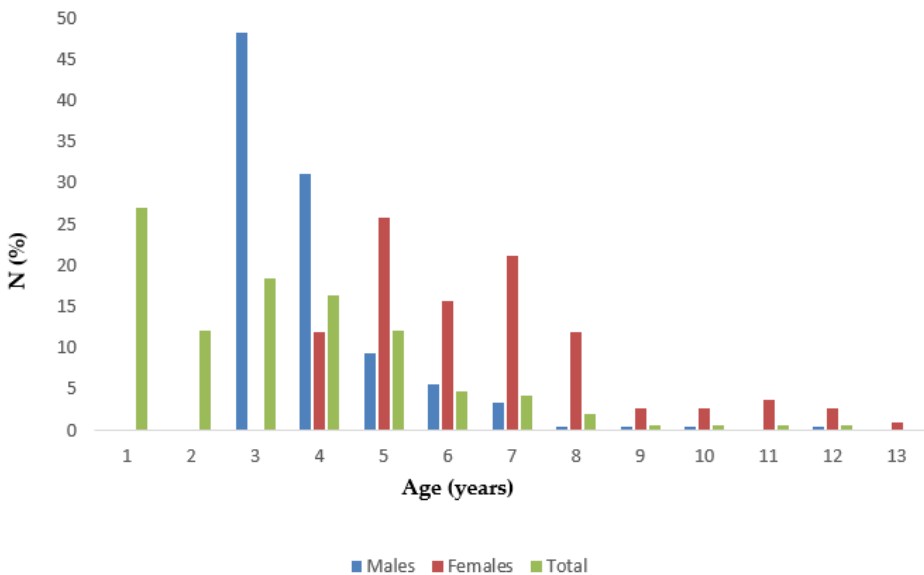

**Figure 4.** Age distributions of the blackspot seabream, *Pagellus bogaraveo,* in the eastern Adriatic Sea.

According to the obtained values of the growth rate (K), it follows that the growth of males was faster than the growth of females (Table 2). The asymptotic value of the length was higher for females (51.70 ± 0.002). The results of the Hotelling $T^2$ test indicated a statistically significant difference in growth parameters between males and females of the blackspot seabream ($T^2 = 221.40 > T_{0\ (0.05,\ 3.16)}^2 = 14.57$), with males growing faster than females. The obtained values of the age at which the blackspot seabream reached 95% of the asymptotic value of the length TL were 14.73 for males, 22.97 for females, and 19.37 for the total sample of blackspot seabream.

**Table 1.** Mean values and standard deviations of total body lengths of males, females, and the total sample of the blackspot seabream, *Pagellus bogaraveo,* in the eastern Adriatic Sea according to age classes. The total sample included males, females, hermaphrodites, and immature individuals, i.e., individuals of indeterminate sex.

| Age Classes | N | Males $\bar{x} \pm$ SD | N | Females $\bar{x} \pm$ SD | N | Total Sample $\bar{x} \pm$ SD | Length Range (cm) |
|---|---|---|---|---|---|---|---|
| 1° | - | - | - | - | 182 | 10.91 ± 1.268 | 8.80–15.10 |
| 2° | - | - | - | - | 81 | 15.49 ± 1.292 | 10.50–17.50 |
| 3° | 88 | 22.72 ± 1.816 | - | - | 124 | 29.91 ± 2.621 | 12.60–28.20 |
| 4° | 56 | 26.90 ± 2.151 | 13 | 28.4 ± 2.084 | 111 | 27.38 ± 2.069 | 18.6 0–31.60 |
| 5° | 17 | 30.34 ± 1.569 | 28 | 30.74 ± 1.376 | 79 | 30.18 ± 1.396 | 27.10–34.10 |
| 6° | 10 | 32.77 ± 1.719 | 17 | 32.61 ± 10.180 | 33 | 32.44 ± 1.324 | 30.70–35.60 |
| 7° | 6 | 33.65 ± 1.185 | 23 | 34.53 ± 1.322 | 28 | 34.35 ± 1.325 | 32.90–36.80 |
| 8° | 1 | 38.20 | 13 | 37.50 ± 1.635 | 14 | 37.54 ± 1.582 | 33.40–39.60 |
| 9° | 1 | 39.60 | 3 | 39.50 ± 0.360 | 4 | 39.53 ± 0.299 | 39.10–39.80 |
| 10° | 1 | 41.20 | 3 | 40.06 ± 0.152 | 4 | 40.35 ± 0.580 | 39.90–41.20 |
| 11° | - | - | 4 | 4158 ± 0732 | 4 | 4158 ± 0732 | 41.00–4260 |
| 12° | 1 | 42.50 | 3 | 4340 ± 0608 | 4 | 4318 ± 0670 | 4250–4380 |
| 13° | - | - | 1 | 4708 ± 0.00 | 1 | 4708 ± 0.00 | 4780 |

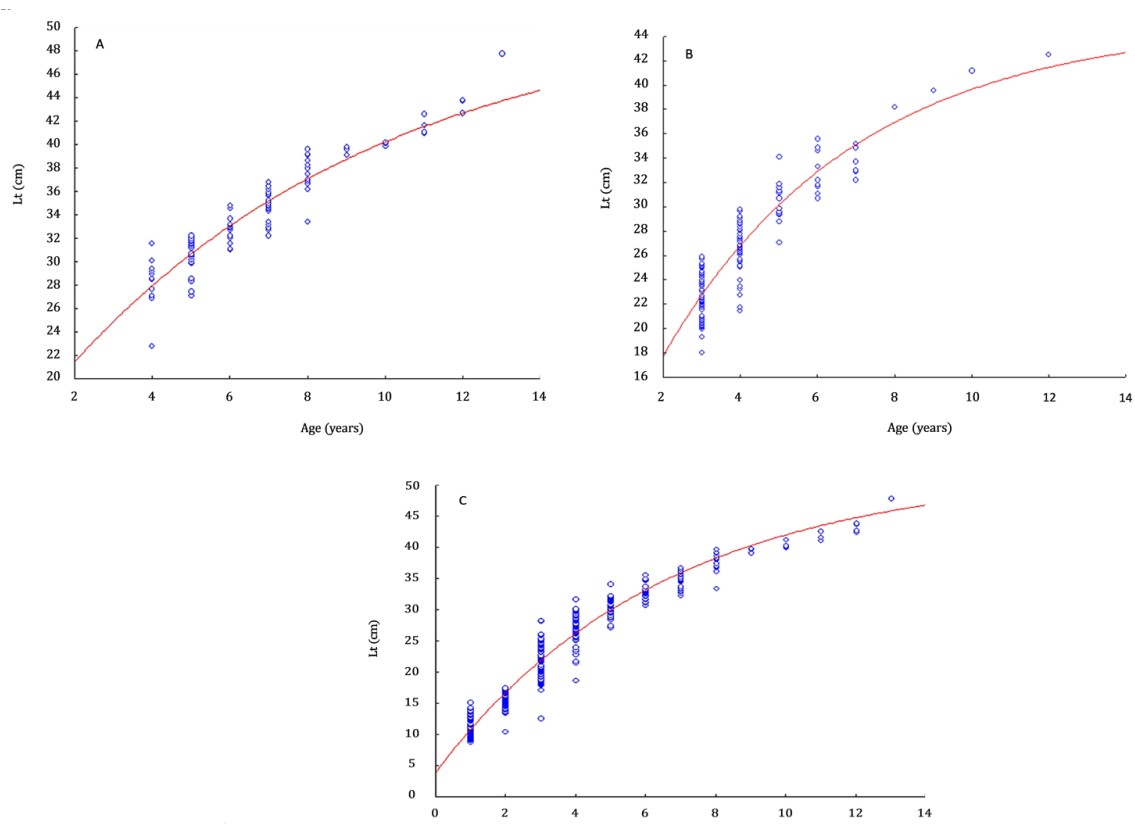

**Figure 5.** Fitted von Bertalanffy growth curve for females (**A**), males (**B**), and total sample (**C**) of the blackspot seabream, *Pagellus bogaraveo,* in the eastern Adriatic Sea (Red line—user specified regression, blue circles—the age at length data).

**Table 2.** Values of growth parameters for males, females, and the total sample of the blackspot seabream, *Pagellus bogaraveo,* in the eastern Adriatic Sea, obtained using von Bertalanffy's growth model ($L_\infty$—asymptotic value of length TL, K—growth coefficient, and $t_0$—theoretical age of the fish at length $L_0$).

| Parameters | Males | Females | Total |
|---|---|---|---|
| $L_\infty$ (cm) | $45.10 \pm 0.002$ | $51.70 \pm 0.002$ | $52.3 \pm 0.0145$ |
| K (years$^{-1}$) | $0.20 \pm 0.009$ | $0.12 \pm 0.005$ | $0.15 \pm 0.002$ |
| $t_0$ (years) | $-0.48 \pm 0.178$ | $-2.42 \pm 0.340$ | $-0.50 \pm 0.031$ |
| $R^2$ | 0.91 | 0.94 | 0.98 |
| $\Phi'$ | 2.60 | 2.50 | 2.61 |

Otoliths of the blackspot seabream are elongated and oval in shape with irregular edges. The anterior region is pointed with a long rostrum. The antirostrum is poorly developed and small. The posterior region is angled obliquely (Figure 6). The otolith morphometry was analyzed on a subsample (N = 230), and the range of total body length of individuals was from 9.30 to 47.80 cm (24.48 ± 10.95). The difference between the mass of the right and left otolith was not statistically significant (*t*-test, *p* > 0.05). The largest range of the obtained values was recorded for otolith length and the smallest for otolith thickness (Table 3). The linear model proved to be the most reliable in presenting the relationship between the total length of the fish and the analyzed otolith parameters (length, width, thickness, and mass) according to the high values of the coefficient of determination (Table 4). The relationships between the observed fish age and otolith morphometrics are presented in Table 5.

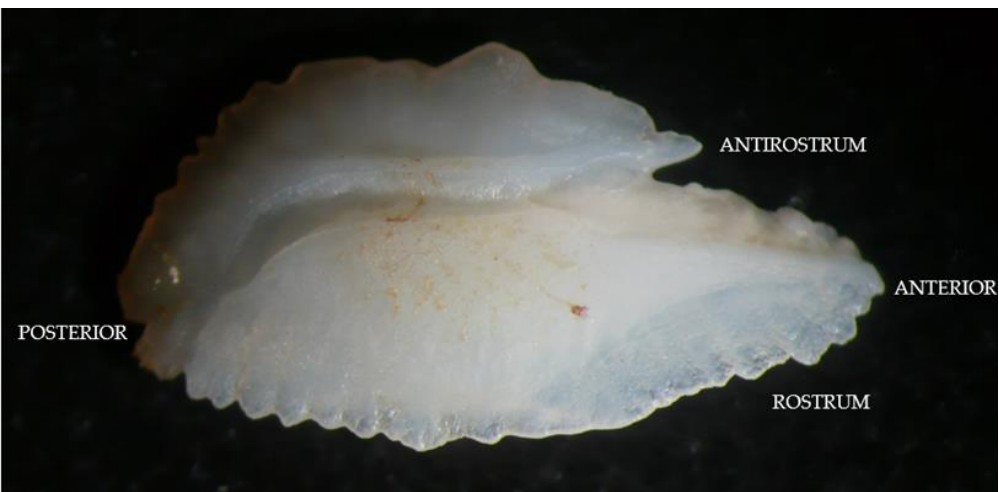

**Figure 6.** Morphology of a sagittal otolith of the blackspot seabream, *P. bogaraveo*.

**Table 3.** Ranges of otolith measurements (OL—otolith length; OW—otolith width; OT—otolith thickness; OM—otolith mass) and their mean values with standard deviation (SD) of the blackspot seabream, *Pagellus bogaraveo,* in the eastern Adriatic Sea.

| Otolith Measures | Range | $\bar{x} \pm$ **SD** |
|---|---|---|
| Otolith length OL (mm) | 3.20–15.00 | $0.87 \pm 0.335$ |
| Otolith width OW (mm) | 1.30–7.00 | $0.49 \pm 0.148$ |
| Otolith thickness OT (mm) | 0.40–2.00 | $0.12 \pm 0.040$ |
| Otolith mass OM (g) | 0.007–0.22 | $0.57 \pm 0.043$ |

**Table 4.** Parameters of the linear regression between fish total length (TL) and otolith morphometrics (OL—otolith length; OW—otolith width; OT—otolith thickness; OM—otolith mass) of the blackspot seabream, *Pagellus bogaraveo*, in the eastern Adriatic Sea (a—intercept value; b—regression slope; $R^2$—coefficient of determination).

| Relationships | a | b | $R^2$ |
|---|---|---|---|
| TL vs. OL | 0.15 | 0.03 | 0.97 |
| TL vs. OW | 0.19 | 0.01 | 0.94 |
| TL vs. OT | 0.03 | 0.003 | 0.86 |
| TL vs. OM | 0.05 | 0.004 | 0.91 |

**Table 5.** Relationships between fish age and otolith morphometrics (OL—otolith length; OW—otolith width; OT—otolith thickness; OM—otolith mass) of the blackspot seabream, *Pagellus bogaraveo*, in the eastern Adriatic Sea, described using the linear model (age = a (otolith morphometric) + b); $R^2$—coefficient of determination.

| Linear Model | $R^2$ |
|---|---|
| Age = 9.52 OL − 3.86 | 0.87 |
| Age = 20.98 OW − 5.84 | 0.74 |
| Age = 74.53 OT − 4.23 | 0.79 |
| Age = 0.0003 OM + 0.04 | 0.02 |

## 4. Discussion

Although this species is widely distributed, there have been no recent studies on the age and growth of this species. Fish growth is an important factor that shapes the population structure of juvenile and adult fish [31,32]. The study of fish age and growth is essential in population dynamics and fishery management [33]. The maximum age estimated in this study by reading otoliths (13 years) was consistent with the maximum ages estimated in the other study areas (Mediterranean and Atlantic areas) [15,17,18]. All studies used the same method and a similar range of total lengths. A higher age (20 years) was recorded only in the Bay of Biscay [32]. The smallest age range was estimated by Mytilineou and Papaconstantinou [12] in the Aegean Sea. The reasons for the age differences could be related to the different temperatures in the environment, the lack of prey, and also the size of the sample [34]. The estimated growth parameters obtained in this study were in line with the available data on the growth and age of blackspot seabream in certain sampling areas (Azores, Cantabrian Sea) (Table 6). Some differences could be a consequence of differing environmental parameters, intrinsic parameters of the population, and sampling bias in various studies.

**Table 6.** Parameters of the von Bertalanffy growth model for different research areas of the blackspot seabream, *Pagellus bogaraveo* (L∞—asymptotic value of length TL, K—growth coefficient, $t_0$—theoretical age of the fish at length $L_0$, $year^{-1}$—unit of measurement for the growth coefficient (K), represents the rate of growth per year.

| Authors | Location | $L_\infty$ (cm) | K ($year^{-1}$) | $t_0$ (years) | $\Phi'$ | Age |
|---|---|---|---|---|---|---|
| Ramos (1967) | Cantabrian Sea | 53.86 | 0.127 | −1.02 | 2.57 | 2–12 |
| Gueguen (1969) | Bay of Biscay | 56.80 | 0.092 | −2.92 | 2.47 | 1–20 |
| Sánchez (1983) | NW Atlantic | 51.56 | 0.209 | −0.53 | 2.74 | 1–12 |
| Krug (1989) | Azores | 58.50 | 0.117 | −1.55 | 2.60 | 1–14 |
| Mytilineou and Papaconstantinou (1995) | Aegean Sea | 25.12 | 0.186 | −2.72 | 2.07 | 0–3 |
| Menezes et al. (2001) | Azores | 56.67 | 0.135 | −1.08 | 2.64 | - |
| Sobrino and Gil (2001) | Gibraltar | 58.00 | 0.169 | −0.67 | 2.75 | 0–8 |
| Chilari et al. (2006) | Ionian Sea | 49.2 | 0.106 | −1.805 | 2.41 | 2–9 |
| This paper | Eastern Adriatic | 52.3 | 0.151 | −0.495 | 2.61 | 1–13 |

Similar values of the growth rate (K) were recorded for the population from the Aegean Sea. The highest asymptotic length ($L_\infty$) was recorded for the population from the Azores area, and the most similar asymptotic value to that obtained in the Adriatic was recorded by the authors in the northwestern Atlantic area [17] and Cantabrian Sea [15]. The obtained values of all the parameters of the von Bertalanffy growth model in this study were within the range of previous studies. The low estimate of $t_0$ can be explained by the lack of individuals smaller than 8.8 cm. The fact that females occupied the largest age groups should also be taken into consideration. A larger number of smaller individuals in the sample would improve age estimates for males and females. Furthermore, a wider age range was recorded in females compared to males. Hermaphrodites were recorded in the age range which corresponded to the time of sex change in the blackspot seabream. The higher asymptotic value of length for females was a consequence of the larger number of females with larger total body lengths in the total sample. Differences in maximum body length for the different parts of the Atlantic and Mediterranean were probably a consequence of the fact that $L_\infty$ is a phenotypic trait of a species and can be limited by different environmental factors. The amount of available food and the sea temperature can affect changes in growth parameters [35]. Larger, older, and late-maturing individuals are often related to colder waters [36]. Elevated temperatures promote premature sexual maturation, and growth slows down or stops completely, so in the end they can result in a shorter length [29]. On the other hand, many organisms grow faster at higher temperatures than at lower ones ("temperature size rule") [37]. Using the parameter Φ', it is possible to compare growth curves between populations of the same species in different areas, but also between different species of the same family [36]. The values of this parameter obtained for the eastern Adriatic population (Φ' = 2.61) were the most similar to the values for the blackspot seabream from the Azores (Φ' = 2.60). The differences in variation can be attributed to environmental conditions affecting population distribution and structure [35].

The analysis of annual rings on sagittal otoliths is one of the most commonly used methods for age estimation [38]. On the other hand, standard otolith reading largely depends on the skill and experience of the person reading them [39]. Accordingly, there have been studies analyzing the relationships between otolith dimensions and fish age [20]. The relationship between otolith dimensions and fish age was described by a linear model that showed that the age of this species in the eastern Adriatic could be best estimated from the length of the otolith ($R^2 = 0.865$), and less so from the mass of the otolith ($R^2 = 0.021$). Lower linear relationships can be associated with an insufficient number of individuals in a particular age group. Some authors suggest that the otolith mass increases more or less constantly throughout the life of the species. A linear relationship between otolith mass and observed age has been found for many species, emphasizing the importance of otolith weight as an age predictor [40]. In most species, the length of the otolith is the best choice because it is a rapid procedure and allows for a slightly better precision than the otolith width. Furthermore, weighing otoliths requires only a precise laboratory scale, and the model has the additional advantage of a relatively constant otolith mass growth rate with age, which could be useful if the sample size range is limited [41]. The otolith morphology of this species from the eastern Adriatic corresponds to the morphological descriptions of this species from the western Mediterranean the north and central eastern Atlantic [27]. The obtained results can be of value in future studies and could also contribute to studies of the feeding habits of various piscivorous species where otolith sizes of prey could be used as a proxy for fish size. Despite its economic importance, *P. bogaraveo* has never been the object of intensive and systematic research in the Adriatic Sea. Therefore, the aim of this paper was to determine the age and growth parameters of this species, which are important input data for stock assessment purposes that form the basis for making fisheries' management decisions. In this sense, given the rising utilization of trawlable deep-water habitats, the significance of such data can be considered even more crucial despite the age of the data. Finally, data presented herein could serve as an important source of data for the management of black seabream fisheries in the Adriatic Sea as well

as in the Mediterranean, especially in the context of climate change and the ever-present over-exploitation of fishery resources.

## 5. Conclusions

The blackspot seabream, *Pagellus bogaraveo*, is a species belonging to the Sparidae family, which is economically important in the Adriatic Sea. Despite its fishery significance, this species has received limited research attention in the Adriatic Sea. This study provides the first comprehensive data on the age and growth of the blackspot seabream in this region. The growth parameters of the species were determined using the Von Bertalanffy model based on age structure as determined by direct otolith readings. The study revealed that the oldest specimen examined was 13 years of age, providing valuable information on the species' longevity in the Adriatic Sea. Regarding the age structure, the study found that males predominantly fell within the 3–4 years age group, while females were more prevalent in the 5–7 years age range. This pattern aligns with the species' protandric hermaphroditism, where individuals change their sex from male to female as they mature. Furthermore, this study reports the first data on the correlation between otolith morphometry and age in blackspot seabream described by a linear model. Notably, there was a strong correlation between otolith length and age, while otolith width and thickness exhibited slightly lower correlations. Surprisingly, otolith mass did not prove to be a significant predictor of age in this species. These findings have important implications and could prove to be valuable in the rapid assessment of age structure in blackspot seabream populations. In conclusion, this study fills a knowledge gap by providing crucial data on the age and growth of the blackspot seabream in the Adriatic Sea and could be particularly valuable for stock assessment purposes, aiding in fisheries' management decisions.

**Author Contributions:** Conceptualization, A.P., N.U., S.M.-S., B.D. and J.D.; methodology, A.P.; software, A.P., N.U., B.D. and S.M.-S.; validation, A.P., N.U., B.D. and S.M.-S.; formal analysis, A.P.; investigation, A.P.; resources, A.P.; data curation, A.P.; writing—original draft preparation, A.P. and N.U.; writing—review and editing, S.M.-S., B.D. and J.D.; visualization, A.P., N.U., S.M.-S., B.D. and J.D.; supervision, A.P., N.U., S.M.-S., B.D. and J.D. All authors have read and agreed to the published version of the manuscript.

**Funding:** This research received no external funding.

**Institutional Review Board Statement:** The study was conducted according to the guidelines of the Code of Ethics of the Faculty of Science University of Split. The study was approved by the Ethics Committee of the Institute of Oceanography and Fisheries in Split on 8 March 2023.

**Data Availability Statement:** Data are available upon reasonable request.

**Conflicts of Interest:** The authors declare no conflict of interest.

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
