# Peer review of "Age, Growth, and Validation of Otolith Morphometrics as Predictors of Age in the Blackspot Seabream, Pagellus bogaraveo, (Brunnich, 1768) from the Eastern Adriatic Sea"

_fishes, doi:10.3390/fishes8060301_

Round 1
Reviewer 1 Report (Previous Reviewer 1)
Dear Authors and Editor
After re-reviewing the revised version of the manuscript, I believe that the Authors have substantially improved it and addressed nearly all of my comments sufficiently. I believe that the new version of the work is definitely better and meets the requirements of the journal.
In my opinion new version of manuscript entitled: Age, growth, and validation of otolith morphometrics as predictors of age in the blackspot seabream Pagellus bogaraveo (Brunnich, 1768) from the eastern Adriatic Sea, should be published in the journal Fishes.
Reviewer 2 Report (Previous Reviewer 2)
All my previous requests were considered during the author's revision. I have no other comments in this regard.
Best regards
The Reviewer
Reviewer 3 Report (Previous Reviewer 3)
The authors did a great job incorporating my original suggestions. The science is sound and the paper will prove useful to readers.
This manuscript is a resubmission of an earlier submission. The following is a list of the peer review reports and author responses from that submission.
Round 1
Reviewer 1 Report
Editorial Office of Journal: Fishes Date: 13.03.2023
Manuscript ID: fishes-2295935
Comments for review of the manuscript entitled: “Age, growth, and validation of otolith morphometrics as predictors of age in the blackspot seabream Pagellus bogaraveo (Brunnich, 1768) from the eastern Adriatic Sea”.
The topics covered in the manuscript are not a novelty in biological sciences and are based on known research methods.
Keywords - in my opinion, the word "otolith" is missing, and the word morphometry has a slightly broader meaning and in this paper should be combined with the word "otolith".
The justification for taking up the topic is in my opinion weak and results only from the fact that we don't know well the biology of this species from the Adriatic Sea. But the biology of this species is known in the Mediterranean Sea. It seems to me that the purpose of the research undertaken should be more justified.
In the paper's aim, the authors indicated that they will compare two methods (lines 48 - 49) and we do not see these two methods in the research methodology.
In my opinion, the Authors should also include parameters such as the fish conservation dimension and the economic size of fish in the study. This would be useful in achieving the goal and importance of this research for more effective management aimed at protecting this species, as indicated in the abstract.
In the introduction chapter, I feel unsatisfied with the current knowledge about the age of fish of this species. What methods of research, what state of knowledge, please describe in more detail, etc.
Materials and methods
I believe that the description of statistical methods should be a separate subchapter in this part of the manuscript. I would also suggest a more detailed description of the tests used, including consideration of experience factors, significance levels, and used software.
There is no explanation of how the "k" parameter (k - growth coefficient) was estimated or derived (line 76)
Results
Due to the imprecise description of the statistical methods in the results chapter, it is not known how the length of male and female individuals (Lines 100 – 101) was compared. It seems to me that such a comparison is inappropriate because it should concern, for example, individual age groups depending on gender, and not entire groups of males and females. And what with hermaphrodites? In addition, the very phenomenon of hermaphrodites in this species requires explanation. The methodological assumptions should precisely describe what the Authors considered male, female, hermaphrodite, and sexually immature individuals.
In my opinion, the notation “(Kolmogorov-Smirnov test; p < 0.01)” (lines 103, 109, 111) also seems to be inconclusive. The Kolmogorov-Smirnov test is about evaluating the data in terms of their distribution and "p < 0.01" is a description of the test statistic in this case Chi-square test. It seems to me that p<0.01 alone would be sufficient. Of course, assuming a correct and comprehensive description of statistical methods in the „Materials and methods” chapter.
The paragraph from lines 119 to 122 should be in the methodological part and not in the results. In addition, in the methodology and/or results, I would suggest including selected photos of otoliths, e.g. from different age groups, e.g. from 3, 8, and 13-year-old fish.
Line 126 - 127 the part of the sentence "... which corresponds to the time of sex change in blackspot seabream." I think it should be used and commented on in the discussion, not in the results.
Line 127 - "The ..." instead of "Tthe ..."
Line 127 - 128 - is this without juveniles? If so, it's worth adding, and there were over 31% of them.
Figure 3 does not bring much new information, as it contains data converted to % from table 1, besides, since the 9th year, the data is not very readable
Table 1 - age class 40 value - with males (6.90??) is it an error?
Shouldn't the (N) males and females sum up to the "Total samples" column? Are the missing individuals hermaphrodites? This should be included in the description of the table.
I would suggest using the same number of decimal places in the values quoted in one table (see the SD column for males and females and, for example, the "Lenght range" column).
Numeric values should be separated by a "." not a ",".
Table 2 seems to be a repetition of the compilation of results from Fig. 1, Tables 2, and Fig. 3. Is it needed?
Why do the Authors not include hermaphrodites and juveniles in the analysis of the results? Perhaps it is worth explaining this in the methodological part.
Figure 4 is not legible enough. I suggest improving it (the font size, removing the yellow background, and others).
Line 147-148 - the sentence "The asymptotic value of length is higher for females (51.7 ± 0.002), which is a consequence of the larger number of females with larger total body lengths in the total sample." I think the second part of this sentence should be placed in the discussion and commented on there.
In my opinion lines, 158 - 162 should be included in the methodology, and here are only the values or photos documenting it.
Line 162 - 164 - since the differences are statistically insignificant, the record of the statistics parameters should be different, i.e.: p > 0.05.
Line 165 - 167 - the sentence should be improved.
Tables 4, 5, and 6 – standardize the notation of decimal places in individual tables. And "." instead "," please.
Line 201 - sentence "Food and temperature are closely related to variations in population parameters [30] seems not correct to me. Shouldn't it be the other way around that population parameters depend on temperature and food availability?
Lines 231 - 233 - I think that such conclusions are excessive because otoliths were collected from dead fish.
I’m not an English native speaker, so I will not rate the language style of the manuscript.
After reviewing this paper I suggest reject of the manuscript in its present form. I believe that after taking into account reviewers’ comments, redrafting the paper, and maybe another review it could be published in Fishes as a research article.
Reviewer 2 Report
I found this manuscript very interesting and accurate about the data analysis and their relapses. The samples, even if not so recent, were well collected and in good quantity to give soundness to the key results of this study. Moreover, the analyses were carried out properly with obvious experience from the research group regarding the topic. I can resume my suggestion for the Authors as follows:
Keywords: Please try to substitute words already reported in the Title with some related ones, to improve the soundness of your manuscript.
Introduction:
Line 35: P. bogaraveo
This section should consider in my opinion also the commercial value of P. bogaraveo in the Adriatic Sea to improve the soundness of the study. Moreover, more information about the literature on the use of Pagellus otoliths and their application should be added.
Results:
Lines 127-128: The eastern. Please double-check the spacing.
Figure 4: Please try to improve the quality, it results very hard to read in the present form.
Conclusion:
Lines 233-237: Please take care to highlight also the limitations of this study, such as the use of samples from about 15 years, when comparing these results with the "management of black seabream fishery".
References:
The references list is in my opinion too short, please try to enrich it with valuable studies on this topic and related to the studies species, especially in the Introduction and Discussion sections, such as:
10.21411/CBM.A.4738CCD6
10.1038/s41598-021-95814-w
10.1093/icesjms/fsq072
10.1111/j.1095-8649.2011.02915.x
10.3989/scimar.2002.66s265
Best regards
The Reviewer
Reviewer 3 Report
This paper which looks at the age and growth of blackspot seabream is very well written and uses appropriate analytical methods to quantify growth and evaluate otolith morphometrics as a predictor of growth. I complement the authors on a straight forward, undistracted attention to important life-history characteristics using classic methods in fisheries science. This is a valuable contribution to the science available for the management of the species. There are a few small suggestions I have made throughout the line be line comments below that should be addressed, but I believe after a minor revision this paper would be suitable for publication in the journal Fishes.
Line by Line Comments:
Line 14: Change commas (,) to periods (.) for all numbers as per English convention. Therefore 8,8 should be 8.8. This should be done throughout the manuscript.
Line 16: Remove ‘the’ before ‘1-year-old’
Line 19-20. The sentence does not make sense as written. Perhaps just say if these morphometric otolith characteristics were suitable predictors of age.
Software that was used to conduct analyses and generate plots should be mentioned.
Line 127: Should be The not Tthe.
Line 48-49: This objective that two ageing methods are used and compared is not actually done in the manuscript based on what I can see. There are no methods or results presented for this objective. Maybe the study changed directions slightly and the goals changed which is completely understandable. I would just remove this objective and that will solve the issue.
Figure 4: The presentation of figure 4 could be improved with use of a different software.
Lines 151-152: Here it says males grow slower than females, but earlier (Line 147) it states that males grow slightly faster than females. The K values suggest males do grow faster than females. This discrepancy needs to be clarified.
Line 164: Should p>0.05 given there left and right otoliths are not significantly different.
Line 183: The “and different life stages” is not necessary, consider removing.
Line 186: ‘other study areas’ instead of ‘most study area’
Line 212-213: Sentence is not clear, consider rephrasing. The point seems to be that increases in temperature can lead to earlier maturity and lower somatic growth. I am not sure this always the case, sometimes elevated temperature increases growth potential.
Line 223: Accordingly not According to.
Line 227-228: I believe there may be other reasons other than low sample size as to why otolith mass is not a good predictor of age. Possibly substantial variation in otolith weight within an age class? Briefly explore other explanations.